# Treatment outcome and associated factors among adult patients with pulmonary tuberculosis in selected health centers in Addis Ababa Ethiopia

**Mehiret Zerihun, Hussen Mekonnen, Tigistu Gebreyohannis Gebretensaye** *

School of Nursing and Midwifery College of Health Science, Addis Ababa University, Addis Ababa, Ethiopia

* tgebreyohannis@yahoo.com, tigistu.gyohannis@aau.edu.et

## Abstract

### Introduction

The success rate of pulmonary tuberculosis in developing countries is different than expected despite effective treatment. We evaluated treatment outcomes and associated factors of pulmonary tuberculosis patients.

### Methods

A retrospective cross-sectional study was employed among randomly selected health centers in Addis Ababa, Ethiopia. Patient records of adult pulmonary tuberculosis patients treated between January 1st, 2017, and December 31st, 2019 were reviewed. Convenient sampling technique was used to select the study participants. Statistical package for social sciences (SPSS), version 24-computer software was used for analysis. Participants' characteristics were descriptively described, and Bivariate, and multivariate logistic regression analysis were used to determine independent variables related to clinical outcomes. The significance level was determined at p-value < 0.05 and a 95% confidence level.

### Results

Six hundred thirty-six patient records with a mean age of 37.49± 2.99 were reviewed. The overall treatment success rate was 84.9%. Absence of comorbid illness [AOR = 0.444; 95% CI:0.219–0.900], non-smoking [AOR = 0.35; 95% CI:0.194–0.645], and being HIV negative [AOR = 0.22; 95% CI: 0.106–0.460] were associated with successful treatment outcomes, whereas, not having treatment supporter [AOR = 15.68; 95% CI: 8.11–30.33] was associated with unsuccessful treatment outcome.

**Data Availability Statement:** The data underlying the results presented in the study are available from the Addis Ababa University repository at

http://etd.aau.edu.et/handle/123456789/24133
and/or the corresponding author on reasonable
request.

**Funding:** The authors received no specific funding
for this work.

**Competing interests:** The authors have declared
that no competing interests exist.

**Abbreviations: AAU**, Addis Ababa University; **AOR**,
Adjusted odds' ratio; **BMI**, Body mass index; **CI**,
Confidence Interval; **DM**, Diabetes Mellitus; **DOTs**,
Direct observation therapy short course; **HCs**,
Health centers; **HIV**, Human immune deficiency
virus; **Kg**, Kilograms; **NTLCP**, National Tuberculosis
and Leprosy Control Program; **OR**, Odd Ratio; **PTB**,
Pulmonary Tuberculosis; **SPSS**, Statistical Package
for Social Sciences; **TB**, Tuberculosis; **WHO**, World
Health Organization.

## Conclusions

Treatment success in this study was below the average target set by WHO. HIV positivity, co-morbidities, and smoking increased risk of treatment failure. Patient education about cessation may improve treatment success.

## Introduction

Tuberculosis (TB) is ranked third among the top ten causes of death from infectious agents worldwide [1]. Asian and African countries share the highest burden of tuberculosis, accounting for about 68% of total cases, which is almost two thirds of the global total [2]. Ethiopia is among the world's 22 highest TB burden countries [3].

Effective treatment and cure requires patients' tolerance and adherence to the full course of treatment period [4]. This is because the microorganism stays inactive for some time and reactivates, making the disease relapse [5]. So being adherent and taking these long term antimicrobial by patients are the major difficulties that may increase risk of unsuccessful treatment outcomes [6]. To maximize the success rate, Ethiopia has been carrying out the directly observed treatment short course (DOTs) strategy since 1992 [7].

TB- HIV co-infection is known to form a lethal combination by speeding up disease progression [8]. Studies reported that treatment failure, death and relapse were mainly associated with HIV infection, Diabetes mellitus, low body mass index [9], and short duration of treatment [6].

According to the WHO 2017 data, the global treatment success rate has reached 85% for new TB cases, 75% for HIV associated, 56% for multi-drug resistant (MDR) and 39% for extensively drug resistant TB [2]. However, the success rate differs from country to country and with the factors that are associated with it [7].

A wide variation in treatment success rate has been seen in European countries. Slovakia and Romania had the highest rate of unsuccessful treatment outcomes, accounting for 66.7 and 55.5% respectively [2]. Unsuccessful treatment outcomes is also a challenge for Sub-Saharan countries like South Africa and Nigeria, where the prevalence of unsuccessful treatment outcomes was 20.4%, and 19.8% respectively [10, 11].

According to the annual performance report of the Federal Ministry of Health of Ethiopia (MOH) in 2017, unsuccessful treatment outcomes showed that 26% in Afar, 22% in Gambella, 18% in Somali and 10% in Addis Ababa [12]. Despite the availability of few studies reporting tuberculosis treatment outcomes among patients with all types of tuberculosis in the country, most of them were hospital based and they failed to examine the risk factors in their reports. Therefore, this study was designed to determine treatment outcomes and associated factors among patients with pulmonary TB in selected health centers in Addis Ababa Ethiopia.

## Methods and materials

### Study design and setting

A retrospective Cross-sectional study was conducted reviewing 694 adult pulmonary TB patient' records who have completed their standard PTB treatment regimen (Rifampicin, Isonized, Pyrazinamide and Ethambutol) in selected health centers in Addis Ababa Ethiopia. Addis Ababa is the capital city of Ethiopia, Headquarter for African Union and United Nations Economic Commission for Africa. It has a subtropical highland climate with an average elevation of 2355 m above sea level with a total surface area of 527 Km$^2$. subdivided into ten sub

cities [13]. The health centers in the city provide preventive as well as curative services to the community under the administration of Addis Ababa health bureau [14].

## Population

Medical records of all pulmonary TB patients treated with standard anti TB drug regimen at the selected health centers in Addis Ababa from January 1$^{st}$ 2017- December 31$^{st}$ 2019 were included.

## Eligibility

All medical records of pulmonary tuberculosis patients aged 18 years and above registered as new, pretreatment and transfer in patients were included. Medical records of pulmonary tuberculosis patients with incomplete data and patients who were transferred to other health facilities were excluded.

## Sampling technique and procedure

Sample size was calculated using single population proportion formula

$$n = Z^2 \times \frac{p(1-p)}{d^2} \text{ DEFF}$$

Where;

n = the required sample size
P = 50% = 0.5 since there is no similar research conducted in Ethiopia
Z = 1.96 (i.e., for a 95% CI)
d = 0.05
DEFF = design effect

$$384*1.5 + 10\% = 633$$

Since the total adult patients treated for pulmonary tuberculosis in the selected health centers were 694, the investigators decided to enroll all charts of pulmonary tuberculosis patients treated in the selected health centers.

Among the ten Sub-cities in Addis Ababa, four sub-cities which include, Nifas silk, Lideta, Gullele and Yeka were selected using simple random sampling method. There were twenty health centers within the four Sub-cities, and (six), health centers were in the same manner the Sub-cities selected. Finally, medical records of all adult pulmonary Tb patients were conveniently included and reviewed.

## Data collection, management and analysis

Structured checklists adopted from previous studies on similar topics were used to extract data [5, 8]. The instrument was pretested in a health center other than the randomly selected health centers. The checklist consisted two sections: (Socio-demographic characteristics and clinical characteristics). Treatment outcomes were categorized as treatment success, when a patient ends up in cure or treatment completion, and unsuccessful treatment outcome, when a patient ends up in treatment failure, death or default. Experienced data collectors were recruited and trained. The principal investigators on a daily basis supervised the data collection process. Data was checked for completeness, coded and entered into Epi-data version 3.5.1 and exported into SPSS version 24 statistical packages for social sciences for analysis. Descriptive statistics were used to describe participant's characteristics, and logistic regression: (Bivariate

and Multivariable) analysis was used to determine association between independent and out-come variables. Variables with a p-value less than 0.2 in Bivariate analysis were included into the multivariable logistic regression model for analysis and a p-value < 0.05 at a 95% confidence interval was considered significant.

### Ethical approval

Ethical approval Ref: AAUMF/02-008/NUR2020 was obtained from the Institutional review board of Collage of Health Science in Addis Ababa University. The need for informed consent was waived by the Institutional Review Committee of the College of Health Sciences, Addis Ababa University, for the fact that, the nature of the study was retrospective. Data was collected by recruiting experienced staff nurses working in the settings, and participants' information was kept private in a safe and secured place to ensure confidentiality.

## Results

### Patient characteristics and treatment outcome of pulmonary TB patients

Among the total of 694 patient charts reviewed, 636 (91.6%) patient records were found to have complete data. The mean age of participants was 37.49± 12.99 with a minimum age of participants 18 years and a maximum of 86 years old. Of these, 361 (56.8%) were male. More than two-thirds, 453 (71.2%) weighed between 55 and 70.9 Kg. The majority, 553 (86.9%) of the participants were Addis Ababa, city residents. More than three quarters, 481 (75.6%) of PTB patients in this study were new cases, 384 (60.4%) of them were smear positive, and 106 (16.7%) were found HIV positive. More than a quarter of the patients, 173 (27.2%) were ciga-rette smokers. Concerning patients' outcomes, the study found that 540 (84.9%) demonstrated successful pulmonary TB treatment outcomes. Of these, 207 (38.3%) were cured and 333 (61.7%) completed treatment. On the other hand, the prevalence of unsuccessful treatment outcomes was 97 (15.2%), of which 18 (2.8%) died during the course of treatment, 33 (5.2%) were treatment failures and 46 (7.2%) were defaulters. Out of the total patients who had unsuc-cessful treatment outcomes, 34% of them were HIV positive, 16.7% were smear positive, and 23% were cigarette smokers. The patient charts review also revealed that, 122 (19.2%) PTB patients were reported to have other co-morbid illnesses, of which 27.9% of them were among the patients who were reported to have unsuccessful treatment outcomes. A total of 87 (13.8%) PTB patient were found to have no treatment supporter, and unsuccessful treatment outcomes were reported among (39.1%) of them (Table 1).

### Treatment outcome among pulmonary tuberculosis patients in selected health centers

The overall success rate of pulmonary tuberculosis patients treated with a standard anti-tuber-culosis treatment regimen was found to be (84.8%).

Treatment success showed incensement with time from (79.4%) in the year 2017 to (92.2%) in the year 2019, and a reduction in mortality rate from 20.4% to 2.04% during the study period. A total of 633 PTB patients' records were reviewed. Patients who end up either in cure or treatment completion were regarded as having successful treatment outcomes, while those who reported treatment failure, death, or default were considered as unsuccessful treatment outcomes (Fig 1).

Data from the study area shows that, considerable proportions (15.2%), of pulmonary tuberculosis patients who took standard anti-tuberculosis treatment regimen have unsuccess-ful treatment outcome.

**Table 1. Patient characteristics and treatment outcome of PTB patients at selected health centers in Addis Ababa, Ethiopia, between January 1, 2017 and December 31, 2019.** (n = 636).

| Characters | | Treatment outcomes | | | | | |
|---|---|---|---|---|---|---|---|
| | Category | Successful | | Unsuccessful | | | Total F (%) |
| | | Cured F (%) | Treatment completed F (%) | Died F (%) | Failed F (%) | Defaulter F (%) | |
| **Age in years** | 18–24 | 32(5.0) | 42(6.6) | 2(0.3) | 7(0.6) | 8(1.25) | 91(14.3) |
| | 25–34 | 73(11.5) | 105(16.5) | 4(0.6) | 5(0.6) | 12(1.9) | 199(31.3) |
| | 35–44 | 63(9.9) | 90(14.1) | 1(0.2) | 9(1.4) | 15(2.4) | 178(28) |
| | 45–54 | 30(4.7) | 63(9.9) | 2(0.3) | 4(0.5) | 7(1.1) | 106(16.7) |
| | ≥55 | 9(1.4) | 33(5.2) | 9(1.4) | 8(1.3) | 4(0.6) | 62(9.7) |
| **Sex** | Male | 105(16.5) | 202(31.8) | 13(2.0) | 7(1.1) | 34(5.3) | 361(56.8) |
| | Female | 106(16.7) | 137(21.5) | 6(0.9) | 14(2.2) | 12(1.9) | 275(43.2) |
| **Residence** | Addis Ababa | 194(30.5) | 290(45.6) | 13(2.0) | 17(2.7) | 39(6.1) | 553(86.9) |
| | Other cities | 17(2.7) | 49(7.7) | 5(0.8) | 4(0.6) | 8(1.3) | 83(13.1) |
| **Weight** | 30–39.9 | 0 | 3(0.5) | 1(0.2) | 0 | 2(0.3) | 6(0.9) |
| | 40–54.9 | 62(9.7) | 60(9.4) | 1(0.2) | 8(1.3) | 11(1.7) | 142(22.3) |
| | 55–70.9 | 144(22.6) | 256(40.3) | 14(2.2) | 12(1.9) | 27(4.2) | 453(71.2) |
| | ≥71 | 5(0.8) | 20(3.1) | 2(0.3) | 1(0.2) | 7(1.1) | 35(5.5) |
| **Patient category** | New | 183(28.9) | 255(40.1) | 9(1.4) | 8(1.25) | 26(4.1) | 481(75.6) |
| | Relapse | 6(0.9) | 23(3.6) | 3(0.5) | 2(0.3) | 5(0.8) | 39(6.1) |
| | Treatment failure | 0 | 9(1.4) | 3(0.5) | 3(0.4) | 9(1.4) | 24(3.8) |
| | Treatment defaulter | 4(0.6) | 15(2.3) | 2(0.3) | 9(1.4) | 2(0.3) | 32(4.9) |
| | Transferred in | 18(2.8) | 37(5.8) | 1(0.2) | 0 | 4(0.6) | 60(9.4) |
| **Other Co morbidities** | Yes | 27(4.2) | 61(10) | 7(1.1) | 12(1.9) | 15(2.3) | 122(19.2) |
| | No | 184(29) | 278(44) | 11(1.7) | 10(1.5) | 31(4.8) | 514(80.8) |
| **HIV/AIDS status** | Positive | 20(3.1) | 50(7.9) | 11(1.7) | 9(1.4) | 16(2.5) | 106(16.7) |
| | Negative | 191(30.0) | 289(45.4) | 9(1.4) | 12(1.9) | 29(4.6) | 530(83.3) |
| **Smear result status** | Positive | 111(17.5) | 98(15.4) | 6(0.9) | 12(1.9) | 24(3.7) | 251(39.5) |
| | Negative | 100(15.7) | 241(37.8) | 12(1.9) | 9(1.4) | 24(3.8) | 384(60.4) |
| **Smoking history** | Yes | 46(7.2) | 88(13.8) | 8(1.3) | 3(0.5) | 29(4.6) | 174(27.2) |
| | No | 165(26) | 251(39.4) | 10(1.5) | 19(3.0) | 17(2.6) | 462(72.7) |
| **Treatment supporter** | Yes | 198(31.1) | 309(49) | 14(2.2) | 8(1.5) | 19(2.9) | 549(86.3) |
| | No | 13(2.0) | 30(4.7) | 4(0.6) | 13(2.0) | 27(4.2) | 87(13.8) |

**HIV**: Human Immunodeficiency Virus, **AIDS**: Acquired Immunodeficiency Syndrome

## Factors associated with treatment outcome among PTB patients

In the logistic regression model, variables with p-value <0.2 in Bivariate regression model were deemed for the multivariate regression models and treatment after failure, treatment after default, other co morbidities, HIV/AIDS serologic status, History of smoking and presence of treatment supporter were found to be the factors significantly associated with TB treatment outcome. TB patients taking treatment after failure were more than 9 times more likely to have unsuccessful treatment outcome compared to new cases and those TB patients taking treatment after default were more than 9 times more likely to have unsuccessful treatment outcome compared to new cases (AOR = 9.105, 95% CI 3.120–26.572) and (AOR = 9.075, 95% CI 3.630–22.685) respectively. Those TB patients who had no other co-morbid condition were 0.44 times less likely to have unsuccessful treatment outcomes as compared to those who have co-morbid conditions (AOR = 0.444 95% CI 0.219–0.90). Similarly, those TB patients with HIV negative

## Treatment outcome

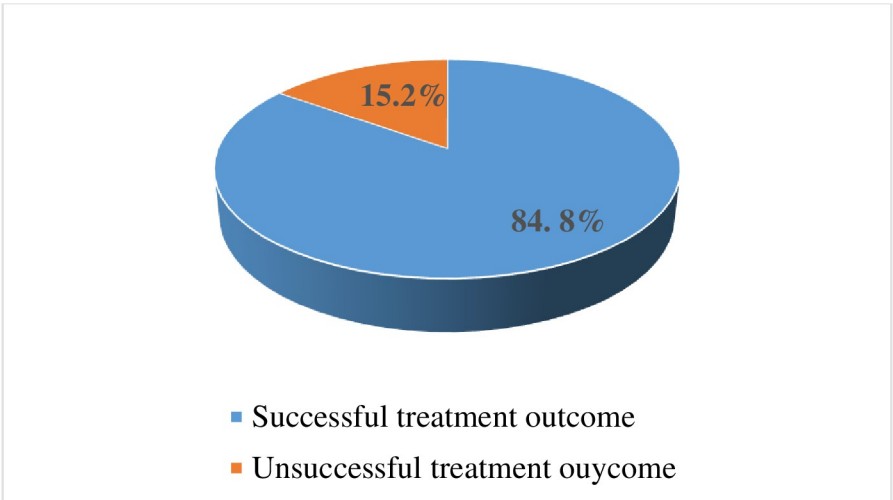

**Fig 1. Treatment outcome among pulmonary tuberculosis patients in selected health centers in Addis Ababa Ethiopia.**

serologic status were 0.22 times less likely to have unsuccessful treatment outcome than HIV positives pulmonary TB patients (AOR = 0.221 95% CI 0.106–0.460). History of smoking was also another determinant factor associated with pulmonary TB treatment outcome. Those patients who had no history of smoking were 0.35 times less likely to have unsuccessful treatment outcomes as compared to those who had history of smoking (AOR = 0.354 95% CI 0.194–0.645). However, lack of treatment supporter was negatively associated with unsuccessful treatment outcome. Those TB patients who were reported to have no treatment supporter were more than 15 times more likely to have unsuccessful treatment outcome than those who reported to have treatment supporter(AOR = 15.684 95% CI 8.111–30.330). (Table 2)

## Discussion

In this study, unsuccessful treatment outcomes among PTB patients were 15.2%, which is higher than the target set by WHO for the year 2020 [2]. The finding was higher than the studies conducted in Jimma and Harar [15, 16], but lower than Gondar, Hosanna, Gambella, and Sodo towns [7, 17–19]. Another study in the Afar region, in Eastern Ethiopia, also reported 18.2%, which is slightly higher than the current study [20]. The success rate has improved between the years from (79.5%) in the year 2017 to (92.2%) in the year 2019 which may be due to improved implementation of the DOTs strategy and improved efforts made to increase patient awareness. The overall mortality rate decreased from 20.4% to 2.04% during the study period, which is consistent with the study done in Denmark [21]. The increase in public health awareness and increased quality of TB control programs may have contributed to the improved outcomes. The current study found that patient re-treatment after failure and/or after defaulters is associated with unsuccessful pulmonary TB treatment outcome. Those treated after failure and/or after default were 9 times more likely to have an unsuccessful treatment outcome than in new cases of pulmonary TB patients. This was similar with findings in Uzbekistan, where a higher proportion of patients encountered unsuccessful treatment outcomes among re-treatment cases [22]. Similarly, other studies [6, 23] also supported the

**Table 2. Factors associated with treatment outcome among TB patients in selected health centers in Addis Ababa, Ethiopia between 1st January 2017 and 31st December 2019.** (n = 636).

| Variable | Category | Treatment outcomes | | AOR | P-value |
|---|---|---|---|---|---|
| | | Successful | Unsuccessful | | |
| **Patient category** | New | 438(69) | 43(6.75) | 1 | |
| | Relapse | 29.0(4.5) | 11(1.8) | 1.804(0.677, 4.811) | 0.238 |
| | Treatment after failure | 9(1.4) | 14(2.2) | 9.105(3.120, 26.572) | <0.0001** |
| | Treatment after default | 19(2.9) | 13(2.0) | 9.075(3.630, 22.685) | <0.0001** |
| | Transferred in | 55(8.6) | 5(0.8) | 0.999(0.339, 2.946) | 0.999 |
| **Other Co morbidities** | Yes | 88(14.2) | 34(5.3) | 1 | |
| | No | 462(73) | 52(8.0) | 0.444(0.219, 900) | 0.024** |
| **HIV/AIDS status** | Positive | 71(11.2) | 35(5.5) | 1 | |
| | Negative | 480(75.4) | 50(7.9) | 0.221(0.106, 0.460) | 0.0001** |
| **History of smoking** | Yes | 135(21.2) | 39(6.2) | 1 | |
| | No | 416(65.4) | 46(7.1) | 0.354(0.194, 0.645) | <0.001** |
| **Treatment supporter** | Yes | 507(80.1) | 42(7.6) | 1 | |
| | No | 43(6.7) | 44(6.8) | 15.684(8.111, 30.33) | <0.0001** |

\* = P-value < 0.05,

\*\* = P-value < 0.01,

AOR (Adjusted odds ratio), 1(reference category), **HIV**: Human Immunodeficiency Virus, **AIDS**: Acquired Immunodeficiency Syndrome

finding by stating that patients who have previously defaulted on their treatment were more likely to fail their treatment compared to their counterparts.

Negative HIV serological test among pulmonary TB patients was another factor significantly associated with positive TB treatment outcome. Pulmonary TB patients with a negative HIV serological test were 0.22 times less likely to have unsuccessful treatment outcome than those pulmonary TB patients who were HIV positive. The finding was similar to the findings of five year retrospective study conducted in the University of Gondar teaching hospital and south-eastern Nigeria [17, 23]. The finding was also supported by another study conducted in Afar regional state in Eastern Ethiopia [20]. The later study also reported that re-treatment of cases has a high chance of having unsuccessful treatment outcomes as compared to new cases like the current study. The finding was also similar to studies at Haramaya University in Eastern Ethiopia and other similar studies [6, 17, 22, 24]. Co-morbid illness among pulmonary TB patients was significantly associated with TB treatment outcomes. In this study, other co-morbid illnesses and smoking were significantly associated with TB treatment outcomes. Pulmonary TB patients who were not smokers were 0.35 times less likely to have unsuccessful treatment outcome than smoking patients, which is consistent with other study findings [25]. Finally, having treatment supporter was found to be significantly associated with unsuccessful TB treatment outcomes. Patients who do not have treatment supporters were more than 15 times more likely to have unsuccessful treatment outcomes compared to those who were reported to have treatment supporters. The finding is similar to the countrywide study in Ethiopia, which reported statistical significance that supporter and degree of drug resistance were linked to TB treatment outcomes [26].

## Study limitations

This facility-based study used in this study was cross-sectional study design, which may hinder to show cause and effect relation-ship. Not all risk factors for TB treatment success were

exhausted, due to incomplete recording in the patient records. The study may have suffered with misclassification bias. The study may be vulnerable for information and selection bias, since diagnostic criteria in health centers may not be based on uniform criteria, making it at risk for information bias, and all cases who don't attend those clinics or seek care during the study period may be missed. Further, it used secondary data in which all conditions together may affect generalizability of the study to a broader community, resulting in drawbacks to draw conclusions beyond the study settings.

## Conclusion and recommendation

Unsuccessful treatment outcomes among PTB patients in this study is 15.1%, higher than the global target set by WHO, lower than 10%. Being HIV negative, not having other co-morbid illness, re-treatment cases (Treatment after failure and Treatment after default), history of smoking were positively associated with treatment success, while, not having treatment supporter was negatively significantly associated with successful TB treatment outcomes among pulmonary TB patients. Patient education, encouragement of drug adherence establishment of home visits and advocating DOTS strategy may be necessary to improve unsuccessful TB treatment outcomes.

## Acknowledgments

First, we would like to acknowledge Addis Ababa University, college of health science school of nursing and midwifery for giving me this chance to do this thesis. I would also like to thank college of health science library.

## Author Contributions

**Conceptualization:** Mehiret Zerihun, Hussen Mekonnen, Tigistu Gebreyohannis Gebretensaye.

**Data curation:** Mehiret Zerihun, Hussen Mekonnen.

**Formal analysis:** Mehiret Zerihun, Hussen Mekonnen.

**Investigation:** Mehiret Zerihun, Tigistu Gebreyohannis Gebretensaye.

**Methodology:** Mehiret Zerihun, Hussen Mekonnen, Tigistu Gebreyohannis Gebretensaye.

**Project administration:** Mehiret Zerihun.

**Supervision:** Hussen Mekonnen, Tigistu Gebreyohannis Gebretensaye.

**Validation:** Hussen Mekonnen, Tigistu Gebreyohannis Gebretensaye.

**Visualization:** Hussen Mekonnen, Tigistu Gebreyohannis Gebretensaye.

**Writing – original draft:** Mehiret Zerihun.

**Writing – review & editing:** Tigistu Gebreyohannis Gebretensaye.

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
