## [Decision Letter · Decision Letter 0]

17 Apr 2023

PONE-D-23-04424Treatment Outcome and Associated factors among Patients with Pulmonary Tuberculosis in Selected Health Centers in Addis Ababa EthiopiaPLOS ONE

Dear Dr. GEBRETENSAYE,

Thank you for submitting your manuscript to PLOS ONE. After careful consideration, we feel that it has merit but does not fully meet PLOS ONE’s publication criteria as it currently stands. Therefore, we invite you to submit a revised version of the manuscript that addresses the points raised during the review process.

We look forward to receiving your revised manuscript.

Kind regards,

Musa Mohammed Ali, PhD

Academic Editor

PLOS ONE

Journal Requirements:

4. Please include a separate caption for figure in your manuscript.

**Additional Editor Comments:**

After careful assessment, your manuscript has merit; however, it does not fully meet PLOS ONE’s publication criteria. Therefore, we invite you to submit a revised version of the manuscript that addresses the points raised during the review process.

1. Give number ‘1’for all authors as they all have the same affiliation. And also correct authors affiliation accordingly “1School of Nursing and Midwifery College of Health Science, Addis Ababa University Addis Ababa, Ethiopia”

2. Is this study retrospective cross-sectional study or retrospective cohort study?

3. Define or describe ‘standard anti TB drug regimen’

4. ‘Sampling technique and procedure’ need revision: how many sub-cities are there in Addis Ababa? “…six health centers were selected based on proportional allocation from each sub city according to the number of health centers in each…” need further elaboration.

5. Ethical approval: “Ethical approval Ref: AAUMF/02-008/NUR2020 was obtained from Research Ethics Committee of Addis Ababa University Collage of Health Science School of Nursing and Midwifery.” Is “……School of Nursing and Midwifery….” Legally registered institution to provide ethical approval?

Reviewers' comments:

Reviewer's Responses to Questions

**Comments to the Author**

1. Is the manuscript technically sound, and do the data support the conclusions?

Reviewer #1: Partly

Reviewer #2: Yes

2. Has the statistical analysis been performed appropriately and rigorously? 

Reviewer #1: Yes

Reviewer #2: Yes

3. Have the authors made all data underlying the findings in their manuscript fully available?

Reviewer #1: Yes

Reviewer #2: No

4. Is the manuscript presented in an intelligible fashion and written in standard English?

Reviewer #1: No

Reviewer #2: Yes

5. Review Comments to the Author

Reviewer #1: I belive that this study will be interesting, but following issues must be solved.

1. The English language used in the submission must be edited.

2. Flow of the article must be corrected.

3. Material and methods section of abstract is over long. No need for such details.

4. The introduction section is also very long. Please shorten this section according to the main objective of your study.

5. You didn’t include the patients below 18 years. Why? If it is possible, please add the pediatric patient, too. If not, the title deserves a revision accordingly. (For example, …among adult Patients with Pulmonary…)

6. You studied a limited number of potential risk factors in your study. Studies reported that diabetes mellitus, drug resistant bacilli, high bacilli load in sputum smear, cavitary lesion in the lungs are the risk factors for unfavorable clinical outcomes of tuberculosis treatment. If it is possible, please consider adding such factors in the study. If not, please discuss them as a limitation.

7. Please extend your discussion with the results of similar studies, weight your findings, and make your suggestions to limit treatment failures.

Reviewer #2: This article talk about the research of the factors associated with the therapeutic success in the tuberculosis treatment in Addis Abeba and around.

In the introduction according to the annual FDRE MOH the success rate in Addis Abeba was 90% in accordance with international requirements but not well understood the inconsistencies because in the same country the reasons can be different by city because different environment.

It would be better to explain the inconsistencies that require this study.

If the term therapeutic success is the desired goal, there must be consistency in the writing to facilitate reading

It will also be necessary to harmonize the font

For the sampling I did not understand the technique of the proportional allocation. This proportionality is a function of the number of patients followed or the geolocalization

6. PLOS authors have the option to publish the peer review history of their article (what does this mean?). If published, this will include your full peer review and any attached files.

Reviewer #1: **Yes: **Yusuf Yakupogullari

Reviewer #2: No

---

## [Author Response · Author response to Decision Letter 0]

9 Jun 2023

Dear Chief-editor and Reviewers,

Thank you for taking your precious time to review our manuscript: ID PONE-D-23-04424 “Treatment Outcome and associated factors among Adult Patients with Pulmonary Tuberculosis in Selected Health Centers in Addis Ababa Ethiopia”.

We have received the reviewer comments from Elsevier’s Author Hub. All comments focused on the instrument section and we have addressed the comments as follows:

Reviewer's comments and their response

Comments: Ensure that your PLOS ONE's style requirements 

Response Manuscript edited to meet PLOS ONE's style requirements

Comment::Include the relevant URLs, DOIs, or accession numbers within your revised cover letter 

Access URLs included in the data availability section and the revised cover letter

Comment: Include a separate caption for the figure in your manuscript

Response: The figure caption is included as recommended

Comment: Give number 1 for all authors as they all have the same affiliation

Response: Changes made as suggested by the editors

Comment: Is this study a retrospective cross-sectional study or a retrospective cohort study?

Response: The study design is a retrospective cohort study since the analysis involves the comparison of groups 

Comment: Define or describe a ‘standard anti-TB drug regimen’

Response: Standard anti-TB drug regimen used by studied patients stated as suggested

Comment: Sampling technique and procedure’ need revision: how many sub-cities are there in Addis Ababa?

Response: Revised and missing components added as per the suggestion 

Comment: State the correct office that is responsible issue ethical approval in the institution.

Response: Ethical approval is issued at the College level and the section is revised as the College of Health Sciences.

Comment: The English language used in the submission must be edited.

Response: The English language was edited in consultation with language experts.

Comment: The flow of the article must be corrected.

Response: The article flow was corrected as per the suggestion.

Comment: The material and methods section abstract is over long

Response: We decided to keep this section since there are no word count recommendations for the journal, and we believed the components are important.

Comment: The introduction section is also very long. Please shorten this section according to the main objective of your study.

Response: Revised as suggested.

Comment: You didn’t include patients below 18 years.

Response: Changes have been made to the title since all participants were adult patients.

Comment: You studied a limited number of potential risk factors in your study.

Response: We were limited to fewer numbers risk factors fact some of the variables were not recorded in the patient records. Therefore, we have included a statement in the study limitation section.

Comment: Please extend your discussion with the results of similar studies.

Response: Efforts were made to make changes in the section

---

## [Decision Letter · Decision Letter 1]

18 Jul 2023

PONE-D-23-04424R1Treatment Outcome and Associated factors among Adult Patients with Pulmonary Tuberculosis in Selected Health Centers in Addis Ababa EthiopiaPLOS ONE

Dear Dr. GEBRETENSAYE,

Thank you for submitting your manuscript to PLOS ONE. After careful consideration, we feel that it has merit but does not fully meet PLOS ONE’s publication criteria as it currently stands. Therefore, we invite you to submit a revised version of the manuscript that addresses the points raised during the review process.

Academic editor’s comment •
Include the sub-heading ‘Introduction’ in  the abstract •
In the manuscript, both retrospective cross-sectional and cohort study designs are mentioned. The appropriate study design needs to be used consistently thorough out the manuscript. •
Paraphrase descriptions in abstract line #35 “Convenient sampling techniques were employed to review patient…”  convenient sampling technique is to be used to select study participants •
Omit “Background” from line #53•
Line #75 “….outcomes was 20.4, and 19.8 respectively” Are these values in % or frequency? •
Line # 123 in a statement “was obtained from Research Ethics Committee of Addis Ababa University..” nomenclature ‘Research Ethics Committee needs to be checked out unless it is changed; the correct name is ‘Institutional Review Board’ check and amends wherever it applies. •
Consider including sample size determination and how proportional allocation was performed?  •
Include all abbreviations used in the tables as footnotes in their expanded form•
Table 1. Are age categories in years? Include unit for weight •
I suggest using ‘multivariable analysis’ instead of ‘multivariate analysis  

We look forward to receiving your revised manuscript.

Kind regards,

Musa Mohammed Ali, PhD

Academic Editor

PLOS ONE

Journal Requirements:

Additional Editor Comments:

Dear Dr. Tigistu.

Thank you for sending your manuscript to PLOS ONE. I have received reviewers’ comments on the revised manuscript; still, there are some points to be addressed before accepting the manuscript for publication. Would you respond to additional comments given by the reviewers’?

Academic editor’s comment

In addition address the following points

• Include the sub-heading ‘Introduction’ in the abstract

• In the manuscript, both retrospective cross-sectional and cohort study designs are mentioned. The appropriate study design needs to be used consistently thorough out the manuscript.

• Paraphrase descriptions in abstract line #35 “Convenient sampling techniques were employed to review patient…” convenient sampling technique is to be used to select study participants

• Omit “Background” from line #53

• Line #75 “….outcomes was 20.4, and 19.8 respectively” Are these values in % or frequency?

• Line # 123 in a statement “was obtained from Research Ethics Committee of Addis Ababa University..” nomenclature ‘Research Ethics Committee needs to be checked out unless it is changed; the correct name is ‘Institutional Review Board’ check and amends wherever it applies.

• Consider including sample size determination and how proportional allocation was performed?

• Include all abbreviations used in the tables as footnotes in their expanded form

• Table 1. Are age categories in years? Include unit for weight

• I suggest using ‘multivariable analysis’ instead of ‘multivariate analysis

Reviewers' comments:

Reviewer's Responses to Questions

**Comments to the Author**

1. If the authors have adequately addressed your comments raised in a previous round of review and you feel that this manuscript is now acceptable for publication, you may indicate that here to bypass the “Comments to the Author” section, enter your conflict of interest statement in the “Confidential to Editor” section, and submit your "Accept" recommendation.

Reviewer #1: (No Response)

Reviewer #2: (No Response)

2. Is the manuscript technically sound, and do the data support the conclusions?

Reviewer #1: (No Response)

Reviewer #2: Partly

3. Has the statistical analysis been performed appropriately and rigorously? 

Reviewer #1: (No Response)

Reviewer #2: Yes

4. Have the authors made all data underlying the findings in their manuscript fully available?

Reviewer #1: (No Response)

Reviewer #2: Yes

5. Is the manuscript presented in an intelligible fashion and written in standard English?

Reviewer #1: (No Response)

Reviewer #2: Yes

6. Review Comments to the Author

Reviewer #1: 1- In the abstract (methods subsection), please shorthen the details of statistical analyses.

2- In the abstract (in results subsection), please use a correct expression for the following sentence: "Comorbid illness

[AOR=0.444; 95% CI:0.219-0.900], smoking [AOR= 0.35; 95% CI:0.194-0.645], being HIV negative [AOR=0.22; 95% CI: 0.106- 0.460], and treatment support [AOR=15.68; 95% CI: 8.11-30.33] were associated with tuberculosis treatment outcome." for better outcome or worst outcome?

3- In the abstract (in results subsection), you stated that the succes rate for your patients was 84.9%; and in Conclusion subsection, you stated that the succes rate found this study was below the WHO's suggestion. You should have evaluated this result according to the patients' risk factors. for example, what was the succes rate among the patients without any risk factor, and what was the succes rate among the patients with one or more risk factor?

I have no further suggestion for the other parts of the manuscript. Good luck for your next studies!

Reviewer #2: I didn't quite understand the term proportionality in this sentence.

« .Within the four Sub-cities, there were twenty health

105 centers and one third (six), health centers were selected proportionally from each sub city

106 according to the number of health centers in each. »

proportional to what?

One of your selection criteria was to receive a selection of centers with a number of people to consult greater than .........

« A structured checklist adopted from previous studies on a similar topic was used to extract data

111 (8) »

The information on the checklist is insuffisant. checklist is validated ? have you done a pretest?

The Authors say that it has been used in several studies but there is only one reference. this checklist should be part of your limits because the measuring tool is not necessarily the same as the studies you are comparing the results of your study.

About the limits of the study

The question is: can this study be generalized to the whole country?

The Authors must add the criteria of information bias with the checklist and perhaps selection bias, as those who were very ill may not have been able to visit the centers.

7. PLOS authors have the option to publish the peer review history of their article (what does this mean?). If published, this will include your full peer review and any attached files.

Reviewer #1: **Yes: **Yusuf YAKUPOGULLARI

Reviewer #2: **Yes: **MAMA DJIMA Mariam

---

## [Author Response · Author response to Decision Letter 1]

26 Jul 2023

Dear Chief-editor and Reviewers,

Thank you for taking your precious time reviewing our manuscript: ID PONE-D-23-04424 “Treatment Outcome and associated factors among Adult Patients with Pulmonary Tuberculosis in Selected Health Centers in Addis Ababa Ethiopia”.

We have received the reviewer comments from Elsevier’s Author Hub. All comments focused on the instrument section and we have addressed the comments as follows:

Response to review comments: 

Additional Information:

S/No Comment Response

1 Include the sub-heading ‘Introduction’ in the abstract 

Sub-heading ‘Introduction’ included in the abstract section

2 In the manuscript, both retrospective cross-sectional and cohort study designs are mentioned. The appropriate study design needs to be used consistently thorough out the manuscript. 

The study design is revised as retrospective cross-sectional study design thorough out the document as recommended.

3 • Paraphrase descriptions in abstract line #35 “Convenient sampling techniques were employed to review patient…” convenient sampling technique is to be used to select study participants 

The statement convenient sampling techniques were employed to review patient charts, paraphrased as suggested in the comment

4 Omit “Background” from line #53

 Sub-heading “Background” omitted from the introduction section.

5 Line #75 “….outcomes was 20.4, and 19.8 respectively” Are these values in % or frequency? T

he outcome values 20.4, and 19.8 are percent and indicated in the mentioned values.

6 Line # 123 in a statement “was obtained from Research Ethics Committee of Addis Ababa University..” nomenclature ‘Research Ethics Committee needs to be checked out unless it is changed; the correct name is ‘Institutional Review Board’ check and amends wherever it applies 

Nomenclature of the Ethics approval body is revised as “Institutional review board of Collage of Health Science in Addis Ababa University”.

7 Consider including sample size determination and how proportional allocation was performed? 

Sample size calculation included as indicated in the appropriate section

8 Include all abbreviations used in the tables as footnotes in their expanded form 

Abbreviations used within tables are indicated as footnotes in their expanded form as recommended 

9 Table 1. Are age categories in years? Include unit for weight 

Unit of measurement for age included as “age in years”.

10 I suggest using ‘multivariable analysis’ instead of ‘multivariate analysis 

Multivariate analysis method is replaced as “multivariable” as recommended by the editor.

 Reviewer 

1 I didn't quite understand the term proportionality in this sentence.

« .Within the four Sub-cities, there were twenty health 

105 centers and one third (six), health centers were selected proportionally from each sub city 

106 according to the number of health centers in each. »

proportional to what?

One of your selection criteria was to receive a selection of centers with a number of people to consult greater than ......... This statement is revised as “Among the ten Sub-cities in Addis Ababa, four sub-cities which include, Nifas silk, Lideta, Gullele and Yeka were selected using simple random sampling method. There were twenty health centers within the four Sub-cities, and (six), health centers were in the same manner the Sub-cities selected”.

2. A structured checklist adopted from previous studies on a similar topic was used to extract data (8). 

The information on the checklist is insufficient. Checklist is validated ? have you done a pretest? 

We have used data from the existing cited sources (5, 8). we assumed these articles to be validated earlier, however the checklist was pretested in a setting other that the selected study settings.

3. The Authors say that it has been used in several studies but there is only one reference. This checklist should be part of your limits because the measuring tool is not necessarily the same as the studies you are comparing the results of your study. 

This is revised and included more citation.

4 Can this study be generalized to the whole country? The response for this question is already reflected in the limitation section, stating “ the findings may not be generalized beyond the study settings.

. 5 The Authors must add the criteria of information bias with the checklist and perhaps selection bias, as those who were very ill may not have been able to visit the centers 

This is also included in the limitation of the study.

---

## [Editor Report · Decision Letter 2]

1 Aug 2023

PONE-D-23-04424R2Treatment Outcome and Associated factors among Adult Patients with Pulmonary Tuberculosis in Selected Health Centers in Addis Ababa EthiopiaPLOS ONE

Dear Dr. GEBRETENSAYE,

Thank you for submitting your manuscript to PLOS ONE. After careful consideration, we feel that it has merit but does not fully meet PLOS ONE’s publication criteria as it currently stands. Therefore, we invite you to submit a revised version of the manuscript that addresses the points raised during the review process.

We look forward to receiving your revised manuscript.

Kind regards,

Musa Mohammed Ali, PhD

Academic Editor

PLOS ONE

Journal Requirements:

Additional Editor Comments:

Dear Dr. Tisgistu,

I have noticed that some of what have been mentioned in point by point response is not reflected in the revised manuscript such as issue of generalization, information and selection bias. All of these are not included in the limitation. Would you revise the manuscript accordingly?

---

## [Author Response · Author response to Decision Letter 2]

2 Aug 2023

Dear Chief-editor and Reviewers,

Thank you for taking your precious time reviewing our manuscript: ID PONE-D-23-04424 “Treatment Outcome and associated factors among Adult Patients with Pulmonary Tuberculosis in Selected Health Centers in Addis Ababa Ethiopia”.

We have received the reviewer comments from Elsevier’s Author Hub. All comments focused on the instrument section and we have addressed the comments as follows:

Response to review comments: 

Upload a copy of Figure 1 which you refer to in your text on page 9

Response; Figure 1 uploaded as separate file

---

## [Decision Letter · Decision Letter 3]

17 Aug 2023

PONE-D-23-04424R3Treatment Outcome and Associated factors among Adult Patients with Pulmonary Tuberculosis in Selected Health Centers in Addis Ababa EthiopiaPLOS ONE

Dear Dr. GEBRETENSAYE,

Thank you for submitting your manuscript to PLOS ONE. After careful consideration, we feel that it has merit but does not fully meet PLOS ONE’s publication criteria as it currently stands. Therefore, we invite you to submit a revised version of the manuscript that addresses the points raised during the review process.

ACADEMIC EDITOR:Dear Dr Tigistu,Some comments given by reviewer 1 were not still addressed. Would you carefully see the comments and revise the manuscript accordingly. I strongly suggest to consider writing style suggested by reviewer 1.

We look forward to receiving your revised manuscript.

Kind regards,

Musa Mohammed Ali, PhD

Academic Editor

PLOS ONE

Journal Requirements:

Reviewers' comments:

Reviewer's Responses to Questions

**Comments to the Author**

1. If the authors have adequately addressed your comments raised in a previous round of review and you feel that this manuscript is now acceptable for publication, you may indicate that here to bypass the “Comments to the Author” section, enter your conflict of interest statement in the “Confidential to Editor” section, and submit your "Accept" recommendation.

Reviewer #1: (No Response)

Reviewer #2: All comments have been addressed

2. Is the manuscript technically sound, and do the data support the conclusions?

Reviewer #1: Yes

Reviewer #2: Partly

3. Has the statistical analysis been performed appropriately and rigorously? 

Reviewer #1: Yes

Reviewer #2: Yes

4. Have the authors made all data underlying the findings in their manuscript fully available?

Reviewer #1: Yes

Reviewer #2: Yes

5. Is the manuscript presented in an intelligible fashion and written in standard English?

Reviewer #1: Yes

Reviewer #2: Yes

6. Review Comments to the Author

Reviewer #1: In the abstract there are still unnecessary details related to statistical analyses. please mention only which statistical analyses were used to assess for which aim. For example, "multivariate logistic regression analyses was used to determine independent variables related to clinical outcomes".

In the abstract, you mentioned the associated factors in the result section but you didn't mention how an associated factor was related to which outcome. For example, HIV negativity was related to treatment succes (or failure)".

Reviewer #2: (No Response)

7. PLOS authors have the option to publish the peer review history of their article (what does this mean?). If published, this will include your full peer review and any attached files.

Reviewer #1: **Yes: **Yusuf YAKUPOGULLARI

Reviewer #2: **Yes: **MARIAM MAMA DJIMA

---

## [Author Response · Author response to Decision Letter 3]

25 Aug 2023

Dear Chief-editor and Reviewers,

Thank you for taking your precious time reviewing our manuscript: ID PONE-D-23-04424R3 “Treatment Outcome and associated factors among Adult Patients with Pulmonary Tuberculosis in Selected Health Centers in Addis Ababa Ethiopia”.

We have received the reviewer comments from Elsevier’s Author Hub. All comments focused on the instrument section and we have addressed the comments as follows:

Response to review comments: 

Comment: In the abstract there are still unnecessary details related to statistical analyses. Please mention only which statistical analyses were used to assess for which aim.

Response: Statistical analysis methods used are corrected and stated based on the recommendation given.

Comment: In the abstract, associated factors mentioned in the result section didn't indicate how an associated factor was related to which outcome.

Response: Correction made to the result section, to indicate which variable is associated to successful, and which ones are linked to unsuccessful treatment outcomes. 

Comment: suggestions on the writing style

Response: Some statements are revised to make sure readers easily understand the paper. Irrelevant statements removed.

---

## [Decision Letter · Decision Letter 4]

18 Sep 2023

Treatment Outcome and Associated factors among Adult Patients with Pulmonary Tuberculosis in Selected Health Centers in Addis Ababa Ethiopia

PONE-D-23-04424R4

Dear Dr. Tigistu,

We’re pleased to inform you that your manuscript has been judged scientifically suitable for publication and will be formally accepted for publication once it meets all outstanding technical requirements.

Kind regards,

Musa Mohammed Ali, PhD

Academic Editor

PLOS ONE

Additional Editor Comments (optional):

Reviewers' comments:

Reviewer's Responses to Questions

**Comments to the Author**

1. If the authors have adequately addressed your comments raised in a previous round of review and you feel that this manuscript is now acceptable for publication, you may indicate that here to bypass the “Comments to the Author” section, enter your conflict of interest statement in the “Confidential to Editor” section, and submit your "Accept" recommendation.

Reviewer #1: All comments have been addressed

Reviewer #2: All comments have been addressed

2. Is the manuscript technically sound, and do the data support the conclusions?

Reviewer #1: Yes

Reviewer #2: Yes

3. Has the statistical analysis been performed appropriately and rigorously? 

Reviewer #1: Yes

Reviewer #2: Yes

4. Have the authors made all data underlying the findings in their manuscript fully available?

Reviewer #1: Yes

Reviewer #2: Yes

5. Is the manuscript presented in an intelligible fashion and written in standard English?

Reviewer #1: Yes

Reviewer #2: Yes

6. Review Comments to the Author

Reviewer #1: Revision suggestions were performed adequately. I think there are some minor polish requirement during the editing.

Reviewer #2: (No Response)

7. PLOS authors have the option to publish the peer review history of their article (what does this mean?). If published, this will include your full peer review and any attached files.

Reviewer #1: **Yes: **Yusuf YAKUPOGULLARI

Reviewer #2: **Yes: **MAMA DJIMA MARIAM

---

## [Editor Report · Acceptance letter]

26 Sep 2023

PONE-D-23-04424R4 

Treatment Outcome and Associated factors among Adult Patients with Pulmonary Tuberculosis in Selected Health Centers in Addis Ababa Ethiopia 

Dear Dr. Gebretensaye:

I'm pleased to inform you that your manuscript has been deemed suitable for publication in PLOS ONE. Congratulations! Your manuscript is now with our production department. 

Kind regards, 

on behalf of

Dr. Musa Mohammed Ali 

Academic Editor

PLOS ONE